# Comparing Super-Resolution Microscopy Techniques to Analyze Chromosomes

**DOI:** 10.3390/ijms22041903

**Published:** 2021-02-14

**Authors:** Ivona Kubalová, Alžběta Němečková, Klaus Weisshart, Eva Hřibová, Veit Schubert

**Affiliations:** 1Leibniz Institute of Plant Genetics and Crop Plant Research (IPK), Gatersleben, D-06466 Seeland, Germany; kubalova@ipk-gatersleben.de; 2Centre of the Region Hana for Biotechnological and Agricultural Research, Institute of Experimental Botany of the Czech Academy of Sciences, 77900 Olomouc, Czech Republic; nemeckova@ueb.cas.cz (A.N.); hribova@ueb.cas.cz (E.H.); 3Carl Zeiss Microscopy GmbH, D-07745 Jena, Germany; klaus.weisshart@zeiss.com

**Keywords:** chromatin, deconvolution microscopy, *Hordeum vulgare*, metaphase chromosome, nanoscopy, photoactivated localization microscopy, stimulated emission depletion microscopy, structured illumination microscopy, topoisomerase II, wide-field microscopy

## Abstract

The importance of fluorescence light microscopy for understanding cellular and sub-cellular structures and functions is undeniable. However, the resolution is limited by light diffraction (~200–250 nm laterally, ~500–700 nm axially). Meanwhile, super-resolution microscopy, such as structured illumination microscopy (SIM), is being applied more and more to overcome this restriction. Instead, super-resolution by stimulated emission depletion (STED) microscopy achieving a resolution of ~50 nm laterally and ~130 nm axially has not yet frequently been applied in plant cell research due to the required specific sample preparation and stable dye staining. Single-molecule localization microscopy (SMLM) including photoactivated localization microscopy (PALM) has not yet been widely used, although this nanoscopic technique allows even the detection of single molecules. In this study, we compared protein imaging within metaphase chromosomes of barley via conventional wide-field and confocal microscopy, and the sub-diffraction methods SIM, STED, and SMLM. The chromosomes were labeled by DAPI (4′,6-diamidino-2-phenylindol), a DNA-specific dye, and with antibodies against topoisomerase IIα (Topo II), a protein important for correct chromatin condensation. Compared to the diffraction-limited methods, the combination of the three different super-resolution imaging techniques delivered tremendous additional insights into the plant chromosome architecture through the achieved increased resolution.

## 1. Introduction

Fluorescent microscopy has become a valuable tool in cell biology research to analyze fluorophore-tagged proteins, DNA, RNA, and their interactions. The resolution obtained by epi-fluorescent microscopy is limited by Abbe’s diffraction limit, defined as the minimum distance between two closely localized structures that can just be distinguished from each other [1]. The achieved resolution in biological specimens is restrained to ~200–250 nm in the lateral plane and ~500–700 nm in the axial dimension [2,3,4]. This limit was overcome by fluorescence nanoscopy, also referred to as super-resolution microscopy. Nowadays, several methods can resolve structures that are below the diffraction limit following different approaches [5]. The first set of these techniques is based on structuring the illumination light obtained by a wide-field configuration resulting in three-dimensional (3D) structured-illumination microscopy (3D-SIM) [6], or by confocal microscopy, resulting in stimulated-emission depletion (STED) microscopy and Airyscan imaging [2,7,8,9]. The second set of nanoscopy techniques works with single-molecule signals of individual fluorophores, later mathematically determined into a final image with ~10–40 nm resolution. To this belong photoactivated localization microscopy (PALM) [10] and stochastic optical reconstruction microscopy (dSTORM) [11,12], commonly referred to as single-molecule localization microscopy (SMLM) [13].

Briefly, in SIM, a pattern, normally a linear grid, is projected into the image plane, which interferes with sample structures creating so-called Moiré fringes. These fringes contain high-frequency information down transformed to low frequencies, which can be captured by the objective lens [2]. Employing a linear grid for structuring, five-phase images have to be taken to restore the down-shifted frequencies to their real position. For a homogenous reconstruction normally three rotations are employed, yielding a total of 15 raw images to reconstruct one SIM image. In linear SIM, a two-fold resolution improvement in all directions can be achieved yielding lateral and axial resolutions in the range of ~100–120 nm and ~350–400 nm, respectively. Fixed material, labeled for conventional microscopy, is well-suited for SIM since it is fully compatible with any commercially available fluorophore. However, the method works best for thin and low-scattering samples as out-of-focus light can pose problems for reconstruction [14,15].

Such a limitation is overcome in Airyscan (Carl Zeiss Microscopy GmbH, Jena, Germany) based on a confocal principle and hence out-of-focus light is efficiently suppressed [16]. In Airyscan the point-spread function itself is used for structuring by projecting the airy disc on a hexagonal detector array. In this way, the airy disc is scanned by the scanning beam as well. The detector elements lying outside the optical axis of the illumination beam contain higher frequency information and reassigning their captured light to the central element, to which their light belongs will result in a super-resolved image. Airyscan can achieve resolutions that come close to SIM in the lateral direction. Since the airy disc is scanned only, a √2 improvement is achieved axially, with a minimal structuring effect.

STED is like Airyscan based on the confocal principle. It acquires images with two spatially arranged laser beams [17]. The first excitation line laser corresponds to the excitation maximum of the fluorophore. The second depletion line laser is engineered into a doughnut shape through a phase mask surrounding the excitation beam. Thus, a small emission spot is recorded while the surrounding fluorescence area is depleted [14,17,18,19]. This corresponds to an almost threefold improvement compared to conventional confocal microscopy. The application to specimens is more limited because the depletion laser acts with very high energy intensity inducing photobleaching and phototoxicity. Consequently, because the theoretical resolution of STED depends on the depletion laser intensity, achieving a higher resolution depends on the sample stability. Hence, very stable bleaching-resistant fluorescence dyes will yield the best results. With those resolutions of ~30–50 nm laterally and ~100–200 nm axially can be reached. Dual-color imaging can be achieved by two dyes depleted with the same laser [2,17].

SMLM relies on the stochastic activation of photo-switchable fluorophores. Switching between the ON (bright; 1st spectral state) and OFF (dark; 2nd spectral state) can be balanced by activation light (normally 405 nm), the power of the illumination light, and the chemical environment (redox cocktail). The goal is to excite only one molecule within the point-spread function (PSF) at a time [20]. In this case, the center of gravity can be determined to a higher precision than the Gauss fit of the PSF would allow. Even the positions of multi-emitters can be accurately fit by special algorithms with minor sacrifices in the localization precision [21]. With such a sparse excitation regime, thousands of images must be recorded and combined into the final SMLM image. In practical terms, SMLM can achieve a higher resolution than STED and SIM, reaching ~20–50 nm laterally and ~10–70 nm axially [2,14]. Practical resolutions of conventional wide-field (WF) and confocal laser scanning microscopy (CSLM) are summarized in Table 1 along with the ones reported by Weisshart et al. [22] for the super-resolution technologies used in this study.

Meanwhile, nanoscopy became essential for plant cell research [15,23,24,25,26,27,28,29]. Choosing the appropriate microscopic techniques is important for the anticipated experiments, and, particularly, for the resolution, they can deliver. To compare the applicability of the different microscopic methods, including super-resolution SIM, STED, and PALM, to investigate chromosome organization, we analyzed the localization of plant topoisomerase IIα (Topo II) in barley metaphase chromosomes within globally DAPI (4′,6-diamidino-2-phenylindol)-labeled chromatin. As the chromosome preparations used in this study were extremely flat, we did not see an advantage to apply Airyscan imaging in addition to SIM. However, the interested reader can find a comparison between Airyscan and SIM to investigate the surface texture and 3D shape of pollen [30], and the Airyscan advantages described when imaging thicker samples [31].

Topo II is a large dimeric enzyme, in humans of ~175 kDa [32]. Its basic function is to introduce DNA double-breaks and thereby resolving catenated or supercoiled DNA molecules in an ATP-dependent manner. Thus, it is important for proper replication, transcription as well as chromosome organization, including mitotic chromosome condensation [33,34]. It was shown that Topo II belongs to the mitotic chromosome scaffolds of vertebrates [35,36,37]. In plants, Topo II is involved in cell cycle regulation as demonstrated for onion [38] and tobacco [39]. Moreover, plant Topo II acts in meiosis to remove bivalent interlocks in Arabidopsis [40]. However, nothing is known about the distribution and function of Topo II in mitotic plant chromosomes. Therefore, we chose this enzyme for our investigations.

In this study, we employed two versatile microscope systems, the Zeiss Elyra PS.1 to perform SIM and PALM/SMLM and the Leica TCS SP8 STED 3X to apply STED. We compared the theoretical and experimentally achieved resolutions for distinct wavelengths generated by the different diffraction-limited and super-resolution techniques and discuss their usefulness to investigate chromosomal substructures and proteins.

## 2. Results

### 2.1. Wide-Field and SIM

The Zeiss Elyria PS.1 microscope system was used to image the chromosomal distribution of anti-barley Topo IIα specific signals. Several distinct Topo II patterns were observed along the metaphase chromosomes. The most eminent signals creating a ring-like structure were detected around centromeres. Another accumulation of Topo II appeared at some (sub)telomers and the nucleolus organizer regions (NOR). Additionally, Topo II appeared without any prominent accumulation dispersed within the chromosome arms. The fluorescence signal intensities, especially at (sub)telomers, varied among the different chromosomes (Figure 1 and Appendix A).

The spatial Topo II arrangement labeled by Alexa488 within DAPI-labeled chromatin was visualized with an increasing resolution by three contrast methods: wide-field, deconvolved wide-field, and SIM. With increasing resolution, also the structural information improved. The resolution of wide-field provided only a general idea about the Topo II localization and chromatin structures. No specific chromosomal substructures could be determined, even not for DAPI-labeled chromatin, for what the short 405 nm wavelength excitation is required. Deconvolution of wide-field removes background noise and improves partially the recognition of two neighboring structures invisible in wide-field. Thereby, network-like chromatin structures became visible. Topo II displayed also more well-defined structures. SIM enhanced further the resolution, consequently, chromatin fibers and small accumulations became detectable. Similarly, the Topo II signals became more distinct, indicating the arrangement of this enzyme into small clusters (Figure 1 and Appendix A). The achieved resolutions according to the Sparrow criterion, compared to the theoretically possible ones, are summarized in Table 2. It becomes evident that the achieved resolutions are excitation wavelength- and microscopy contrast mode-dependent. The resolution of STAR635P-labeled Topo II imaged by SIM showed also more details than wide-field and deconvolved wide-field, but it was less than for Alexa488-labeled Topo II (Figure 2 and Table 2). 

### 2.2. Confocal and STED Microscopy

Next, the chromatin structure and Topo II distribution were analyzed by super-resolution microscopy using a Leica TCS SP8 STED 3X confocal microscope. Similar to deconvolved wide-field microscopy, confocal microscopy showed the network-like chromatin structures labeled by DAPI during the flow-sorting of the chromosomes. Due to the properties of the microscope (equipped with a 775 nm depletion laser), STED microscopy can be performed with fluorescence dyes which can be excited at 580 and 635 nm wavelengths. Consequently, chromatin visualizations stained with DAPI cannot be performed by STED. The confocal imaging confirmed the Topo II localization pattern found by spatial wide-field and SIM (Figure 3 and Appendix A). Applying STED microscopy delivering a resolution of ~134 nm revealed more precise structures than SIM (~171 nm resolution). Post-processing of STED images through deconvolution to reduce blur resolved ~72 nm, as measured according to Sparrow [41]. The achieved resolution corresponds to the calibration with 80 nm gold beads. No resolution differences were observed in the two mounting media applied (hardening Diamond and non-hardening DABCO) (Table 3). Because imaging of chromatin (labeled with DAPI) by STED is impossible the confocal DAPI and STED images were merged to colocalize the chromatin and Topo II patterns (Figure 3 and Appendix A).

### 2.3. PALM

Since PALM is the official product name of the ELYRA PS.1 system, we use this term instead of SMLM. 3D-PALM analyses resulted in similar chromosomal Topo II patterns in the chromosomes as described above. Moreover, even single Topo II molecules forming clusters were identified. The minimal peak-to-peak distance for centroids within the cluster was measured with ~5 nm (Figure 4 and Appendix A). Please note that the distance of centroids is not related to the localization precision, which is the standard deviation signifying at which certainty a peak was localized. The median of the localization precision can be extracted either from the histogram as the number with the highest absolute frequency (Appendix A), or directly from boxplots (Appendix A). In the shown example the majority of molecules detected by the Topo II antibodies raised in rabbit (rb12) were localized with a mean precision of 55.1 nm laterally, with 78.2% of the molecules falling into a range of 10–80 nm (Figure 4). The median of the axial localization precision amounted to 56.9 nm, with 77.8% of the molecules falling into a range of 10–80 nm (Appendix A). It should be emphasized that the localization precision determines the maximal possible resolution that will only be present if the distance between two neighboring fluorophores is at least half the localization precision. If that is not the case, the resolution will be just twice that spacing. Similarly, using the Topo II antibodies raised in guinea pigs (gp13), the Topo II molecules were localized within whole chromosomes and their pericentromeric regions inside, and the lateral and axial resolutions were determined (Appendix A). The obtained medians were taken to derive the distribution (boxplots) and mean values. The lateral precisions of antibodies raised in the different animals were ~52–54 nm and ~45–46 nm for chromosomes and centromeric regions, respectively. Since the data were obtained from the same measurements, the higher precision obtained for pericentromers might result from better access of that region for the reducing agent (β-mercaptoethanol) or a higher quantum yield. The axial resolutions were ~50–60 nm for the chromosome and ~45–55 nm for the centromeric region. In general, the rabbit-derived antibodies yielded slightly better localization precisions, which might be due to a higher affinity of these antibodies, or better steric access to the conjugated dye. The similar precisions obtained with two different primary Topo II antibodies (rb12 and gp13) demonstrates the reliability of the PALM method.

The Elyria PS.1 microscope system permits performing wide-field, deconvolution, and SIM in parallel to get a structural specimen overview. Afterward, PALM at the identical specimen reaches the resolution of standard electron microscopy. The combination of spatial SIM and PALM increases significantly the understanding of the ultrastructures by distinct molecule localization. It has to be emphasized that the ring-like centromeric structure appearing to be mainly closed in WF, DCV, SIM and STED occurs more open in the PALM localization map. We consider this as real because in the DCV and SIM images such a gap is indicated as well (Figure 4a).

## 3. Discussion

The image resolution is important for assessing biological systems. The development of light microscopy from diffraction-limited wide-field via highly resolved confocal microscopy up to super-resolution techniques enabled the exploration of new structural features through the localization and co-localization of nucleic acids and proteins. To identify higher-order chromatin structures, such as 100–300 nm chromatin fiber loops, a resolution at the border of the diffraction limit is required. Going deeper into the chromatin structure, nucleosome clusters must be detected. These ultra-structures can be identified by SIM and STED microscopy [24]. According to electron microscopy imaging, the diameter of a Topo II dimer is ~15 nm [42], thus in praxis, only SMLM would be able to separate such clusters if in such close proximity.

In our study, we applied three nanoscopic techniques at chromosomes of barley and confirmed their ability to image chromatin and proteins such as Topo II in the expected resolution ranges. However, as commercial systems do not use the full numerical aperture of the objective, the theoretically possible resolution of SIM could not be completely achieved. Moreover, SIM uses in image reconstruction a regularization step called the Wiener filter, or less accurately Wiener deconvolution. This Wiener filter can be set at different strengths which determines at which point high frequencies are cut off. The higher the frequencies one wants to capture corresponding to a better resolution, the less stringent that filter has to be set. However, if there is much noise in the image, which represents the highest frequencies, setting the filter too soft will result in structured noise that will be shown as honeycomb structures. This will also afflict the true structure under investigation. Hence, the filter setting has to be compromised to avoid such artifacts.

The reached resolution by STED, equipped with a 775 nm far-red laser was ~134 nm. This limit is set by the signal-to-noise in the image. The stronger the depletion laser power is set the less signal will be obtained from the smaller illumination spot. The resolution reported here is similar to the SIM resolution achieved by the 405 nm laser excitation. This wavelength is usually used to excite chromatin counterstaining dyes, like DAPI. For STED microscopy equipped with a 775 nm laser chromatin counterstaining with e.g., DAPI and Hoechst 33,342 cannot be applied. For STED, dyes shifted towards the red spectrum are generally preferred to minimize autofluorescence during imaging. Therefore, dyes with fluorescence excitation within the blue and green spectral ranges are avoided. To achieve the best-resolved chromatin with low background, Spirochrome’s SiR probes emitting light in the near-infrared spectral range are recommended.

Moreover, DAPI and Hoechst 33,342 might have a negative influence on the image quality. Due to the crosstalk between DAPI signals and signals acquired by the depletion laser (especially with the 592 nm laser) background noise may occur. By applying deconvolution to the STED images, we achieved a final resolution of ~72 nm. The theoretical resolution of a commercial system with STED-optimized dyes can reach a ~50–60 nm lateral resolution [8]. As in SIM, the enhancement in resolution by the deconvolution step will be limited by the strength of the noise filter that can be set without creating artifacts. The achieved resolution also depends on the preparation method and the calibration beads used to precisely calibrate the STED imaging. The beads have to be selected based on the theoretical size of the acquired molecule complexes, and the depletion laser intensity. Increasing the intensity of the depletion beam can improve the resolution. Unfortunately, high depletion laser power can provide photobleaching and phototoxicity of biological samples. Thus, the power of the depletion laser must be optimized during the acquisition for the utilized dyes to get the highest possible resolution without bleaching.

Employing SMLM, individual Topo II molecules can be distinguished and hence counted. For best resolution the density and blinking ability of single molecules is crucial. The majority of single molecules displayed an XY-precision/resolution of ~40–60 nm, with few molecules localized with 10–20 nm precisions. In our study, this limit of potential resolution is mainly caused by the fact that the high molecule density prevented single molecules per PSF to be in the ‘on’ state. Additionally, fitting multi-emitter events will always induce a loss in localization precision. Overall, the findings in this study agree with the previous works in Arabidopsis to co-localize single RNA polymerase II molecule variants in interphase nuclei [28] and to localize plant-specific END BINDING1c (EB1c) proteins, members of the microtubule plus end-binding protein family of Arabidopsis, in root epidermal cells [23]. Similarly, dual-color 3D-dSTORM was applied to co-localize ROXY1, a plant-specific glutaredoxin, and distinct RNA polymerase II molecules throughout the transcription cycle in Arabidopsis root meristem nuclei [25]. Thus, SMLM, including PALM and dSTORM, proved to be very powerful to identify and localize accurately single molecules and to corroborate data obtained from SIM.

Besides the specific microscopical method applied, the achievable resolution depends on sample preparation as well. While SIM works completely with all samples prepared for wide-field microscopy, i.e., with standard dyes, STED is more challenging. Until now, in plant cell research STED microscopy used mostly only one far-red (775 nm) depletion laser with a two-color acquisition possibility (data not shown). Recently, new specific STED-optimized dyes were developed to enable the acquisition of up to three colors using one far-red STED depletion laser (Leica microsystems guide for STED sample preparation). Moreover, STED microscopes equipped with two depletion lasers (775 nm and 595 nm) were developed to acquire additional fluorescent dyes. However, the calibration of both depletion lasers can be quite challenging [43].

For STED the specimen becomes mounted on coverslips to manage the shortest working distance and is, ideally, labeled with specific photo-stable dyes (Abberior company) to minimize bleaching caused by the high power of the depletion laser. SMLM also requires the sample preparation onto coverslips and not slides, because the imaging has to be performed in a coverslip chamber containing α-mercaptoethanol in 1× PBS to induce molecule blinking. The inclusion of the structure of interest by cytoplasm can impair PALM, especially if the chemical redox cocktail is hampered in its access to the fluorophore. Consequently, flow-sorted nuclei [15,44] and isolated chromosomes (this study) free of cytoplasm are much better suited for this approach. Generally, also the background noise is lower when working with isolated organelles.

Therefore, the decision of which visualization method to choose depends on several factors including the aim of the study, the sample preparation method, the required sample handling and staining, dye properties, and the required resolution. SIM is well suited when a fast co-localization of two or more proteins/DNA/RNA is required. Because SIM involves only the linear interaction of lower laser power light with the sample, it is the most quantitative and life cell compatible method among the super-resolution techniques. PALM is the method of choice to analyze molecule clusters at the highest resolution, and if molecule counting is of interest. STED becomes handy if resolutions more than two-fold are required, and as a confocal technique, it can handle thicker samples.

Taken together, while conventional wide-field microscopes may deliver a rough overview of chromatin organization and protein localization, wide-field deconvolution and confocal microscopy increase the visibility of structures at diffraction-limited resolution. The super- resolutions achieved by SIM and especially STED show more structural details. The highest resolution can be obtained by PALM even allowing single-molecule localization. The combination of SIM and PALM is quite useful to show an ultrastructural overview with single molecules distinctly localized inside of these structures.

## 4. Materials and Methods

### 4.1. Plant Material and Specimen Preparation

Chromosomes of barley (*Hordeum vulgare* L. cv. Morex) were sorted according to Lysák et al. (1999) [45]. Briefly, suspensions of mitotic metaphase chromosomes were prepared from synchronized root tips of ~3 cm long primary roots. DAPI-stained chromosomes were analyzed and flow-sorted by FACSAria II SORP flow cytometer and sorter (BD Bioscience, San Jose, CA, USA). Five thousand chromosomes were sorted onto high precision cover glasses (Paul Marienfeld GmbH and Co. KG, Lauda-Königshofen, Germany) into 15 μL of PRINS buffer supplemented with 2.5% sucrose (10 mM TRIS, 50 mM KCl, 2 mM MgCl_2_.6H_2_O, 2.5% sucrose; pH 8.0). The slides with sorted chromosomes were stored at room temperature (RT) overnight and immuno-labeled the next day, or stored at −20 °C for a longer period.

To confirm the specificity of the Topo IIα antibodies by peptide competition, meiotic cells were prepared from barley anthers using the squashing method. Briefly, anthers were fixed in 4% formaldehyde (#18814-10, Polysciences, Warrington, PA, USA) in dH_2_O on ice for 50 min under vacuum, and then treated with an enzyme mixture (0.1% Pectolyase (Sigma); 0.1% Cytohelicase (Sigma); 0.07% Cellulase R-10 (Duchefa), and 0.07% Cellulase (Calbiochem) in 1 × PBS) for 45 min at 37 °C. Subsequently, 3–4 anthers were dissected in a drop of distilled water, covered by a coverslip, and frozen in liquid nitrogen. After coverslip removal, the slides were immersed into phosphate buffer saline (1 × PBS; 10 mM Na_2_HPO_4_, 2 mM KH_2_PO_4_, 137 mM NaCl, 2.7 mM KCl, pH 7.4) and used for immunostaining.

### 4.2. Topo II Antibody Preparation and Specificity Proof

Topo IIα antibodies (Topo IIrb12, Topo IIgp13) were raised by the LifeTein company (Somerset, NJ, USA) in rabbits and guinea pigs against the synthetic peptides GDAAGKTIEEMYQKKTQLE-C and C-DEDIAEPQHESEDEGSSME, respectively, of the barley Topo IIα gene HORVU6Hr1G067930. Purified polyclonal antibodies were diluted in 1 × PBS with 0.02% sodium azide to obtain a stock solution of 4.6 mg/mL or 5 mg/mL.

To prove the specificity of the Topo II antibodies by peptide competition, the specific peptides against Topo II used for the rabbit and guinea pig immunization was reconstituted in 1 × PBS with 0.02% sodium azide to obtain a stock solution of 5 mg/mL. The Topo IIrb12 and Topo IIgp13 antibodies were mixed with the peptides in antibody solution to the final dilution of 1:100 of antibodies. The peptide concentration used for Topo IIrb12 was 1:100, 1:20, and 1:10, and 1:100 and 1:50 for Topo IIgp13. The mixture was incubated overnight at 4 °C. The next day, the slides with sorted chromosomes and meiotic tissues were blocked as described above and incubated with the antibodies/peptide mixture overnight at 4 °C. Afterward, the slides were washed in 1 × PBS and incubated with secondary donkey anti-rabbit Alexa488 antibodies (1:200, #711-545-152 Jackson ImmunoResearch, West Grove, PA, USA), and diluted in antibody solution for 1 h at 37 °C. Next, the slides were washed three times in 1 × PBS at RT followed by dehydration in an ethanol gradient (70%, 85%, and 100%), each step 1 min. Air-dried slides were counterstained with DAPI and subjected to microscopy (Appendix A).

### 4.3. Indirect Topo II Immunostaining for the Zeiss Elyra PS.1 Microscope System

Slides with sorted chromosomes were blocked using a blocking solution (5% BSA, 0.03% Triton X-100, 1× PBS) for 1 h at RT and incubated with primary antibodies against Topo II diluted (1:100) in antibody solution (1% BSA, 0.01% Triton X-100, 1 × PBS) overnight at 4 °C. Next, after washing in 1 × PBS, slides were incubated with secondary donkey anti-rabbit Alexa488 antibodies (1:200, #711-545-152 Jackson ImmunoResearch), diluted in antibody solution, for 1 h at 37 °C. Subsequently, the slides were washed in 1 × PBS at RT and immediately dehydrated in an ethanol gradient (70%, 85%, and 100%), each step 1 min. Next, slides were air-dried, counterstained with 1 µg/mL DAPI in antifade (Vectashield, Vector Laboratories, Burlingame, CA, USA), and subjected to microscopy.

### 4.4. Indirect Topo II Immunostaining for the Leica TCD SP8 STED 3X Microscope

Slides with sorted chromosomes were incubated in blocking buffer (3% BSA, 0.5 M EDTA, Tween 20) for 1 h at RT followed by incubation with Topo IIrb12 primary antibodies (diluted 1:125) overnight at 4 °C. Afterward, slides were washed in 1 × PBS at RT and incubated with secondary anti-rabbit antibodies STAR 635P (ST635-1002-500UG, Abberior, Göttingen, Germany), diluted 1:125 in blocking buffer, for 1 h at RT. Next, slides were washed in 1 × PBS at RT, dehydrated in an ethanol gradient (70%, 90%, 100%), and mounted in two different embedding media. Samples embedded in the ProLong diamond medium (ThermoFisher Scientific/Invitrogen, Waltham, MA, USA) were cured for 24 h at room temperature in the dark before microscopy. Slides with DABCO mounting medium (Sigma-Aldrich, Darmstadt, Germany) were sealed with Fixogum rubber cement (Marabu GmbH and Co. KG, Tamm, Germany) to prevent evaporation and stored at 4 °C before microscopy.

### 4.5. Wide-Field, Deconvolution, 3D-SIM, and PALM

The fluorescence signals of Topo II were imaged by wide-field (WF), deconvolution (DCV) of WF, and super-resolution 3D-SIM, using an Elyra PS.1 microscope system equipped with a 63×/1.4 Oil Plan-Apochromat objective and the software ZENBlack (Carl Zeiss GmbH, Jena, Germany). Images were captured separately for DAPI and Alexa488 using the 405 nm and 488 lasers for excitation and appropriate emission filters [44]. Ca. 20 slices were captured within a ~2 µm Z-stack. Reconstruction of SIM images was done with the ZENBlack software structured illumination processing module. Wiener deconvolution was used as implemented in the Zeiss SIM module [46]. In a first step, the Wiener filter was set free and automatically determined, followed by a systematic alteration of its value. The minimal strength, where just no structured noise was visible, was selected. PALM was also performed with the 405 nm and 488 lasers [44]. PALM images were processed with the ZENBlack software PALM processing module. Routinely method-based drift correction was performed, and grouping was applied. Localization events were fitted by a Gauss function. Either centroids or the Gauss function was plotted. 3D rendering of SIM and PALM image stacks to produce movies was performed with the Imaris 9.6 software (Bitplane AG, Zurich, Switzerland).

### 4.6. Confocal and STED Microscopy

Images were acquired using a Leica TCS SP8 STED 3X confocal microscope (Leica Microsystems, Wetzlar, Germany), equipped with an HC PL APO CS2 100×/1.44 Oil objective, Hybrid detectors (HyD), and the Leica Application Suite X (LAS-X) software version 3.5.5 with the Leica Lightning module (Leica, Buffalo Grove, IL, USA). Confocal image stacks of chromosomes were captured separately in sequential scans, to avoid spectral mixing, using 635 nm and 405 nm laser lines for excitation and appropriate emission spectrum. For Z-stack imaging, the confocal pinhole was set to 1 AU, and the Z-step size was set to ‘system optimized’ to avoid under-sampling. Ca. 8–12 slices with a distance of 0.2 µm were captured within a Z-stack. To achieve super-resolution, STED beam alignment was performed between the white light laser and the 775 nm depletion STED laser before imaging. The spatial alignment of the laser beam was performed using gold beads with a diameter of 80 nm (Sigma-Aldrich, Darmstadt, Germany), before image scanning. The point spread function was visualized by recording the backscattered light from the 80 nm gold beads illuminated with the white light laser source and the STED laser (775 nm). The pixel size of the acquisition was applied automatically in the LAS-X software and resulted in a final value of less than 20 × 20 nm. Images were captured using a pixel dwell time of 100 ns. The power of the depletion laser was optimized during the acquisition of the STAR 635P dye to get the highest resolution without bleaching. Photon time gating was used to collect lifetimes between 0.3 and 6 ns. The hybrid detector gain was 100%. All pictures were captured in standard mode and deconvolved with the Leica Lightning module using pre-settings based on the refractive index (1.4615 for DABCO and 1.46 for Diamond) of the mounting medium. For confocal image deconvolution, the following parameters were used: strategy—adaptive; the number of interaction—auto; contrast enhancement—auto; regularization method—goods roughness; regulation parameter—0.05; optimization—very high; post filter—none. For STED image deconvolution, the following parameters were used: strategy—adaptive; the number of interaction—auto; contrast enhancement—auto; regularization method—goods roughness; regulation parameter—0.05; optimization—very high; post filter—none. Image processing, including conversion of imaged Z-stacks into maximum intensity projections (MIPs), was performed with the LAS-X software, and the final images were arranged in Adobe Photoshop software version 6.0 (Adobe Systems Corporation, San Jose, CA, USA).

### 4.7. Resolution Considerations

Resolution is defined as the ability to discriminate two structures in the image. Resolution is not absolute as there exists more than one definition [47]. The common criteria for fluorescence self-emitters are listed in Table 4.

The Raleigh criterium states, that in order that two emitters are resolved, the maximum of the intensity distribution (the Airy disc in the lateral direction) of one emitter must fall in the first null of the one of the other emitter. For the full width at half maximum (FWHM), the Gauss intensity distributions of two emitters must not overlap above the 50% maximum value. For Sparrow, only a detectable decrease in brightness between the two maxima has to be present.

In confocal and super-resolution microscopy, these resolutions can be improved by certain factors that depend on the technology [22] and which are summarized in Table 5.

Equations in Table 5 predict unlimited resolution for those microscopy techniques that draw on saturation effects (STED and SMLM). There is as well a practical limit set by the signal-to-noise or photon budget for these technologies.

To achieve a certain resolution, the spacing between neighboring fluorophores must be at least twice as fine according to the Nyquist theorem [48], or in other words, their distance needs to be at least half of the resolution. Therefore, also label size will matter. For example, if staining with an antibody of 10 nm in size is used, the maximum resolution will be twice this size, namely 20 nm. This can be especially significant for SMLM, for which the localization precision defines the maximum resolution achievable. If, however, spacing does not meet the Nyquist criterion, the resolution will be just twice the distance of the fluorophores. Unfortunately, labeling densities are normally not known and are hard to measure. They might also be quite different at different sites in the image and accordingly will be the resolution.

As resolution maps are hard or nearly impossible to obtain, one must resort to drawing profiles along the structures (Appendix A). The easiest criterion to measure resolution in this way is according to Sparrow [41], as only a dip to 98.6% of the maximum is required. FWHM and the Raleigh criteria would need deeper dips to 50% and 37% between the maxima, respectively (Table 6), which would require fitting the profiles to Gauss distributions.

### 4.8. Resolution Measurements

Resolution measurements in wide-field, deconvolved wide-field, and SIM images acquired by the Elyra PS.1 microscope was calculated according to Sparrow [41], where the resolution is defined as the minimal distinguishable distance between the intensity maximums of two structures [49] (Figure 5). To use this approach for PALM the Gauss display must be used, as distances between centroids are not related to localization precision and hence resolution. Similarly, the resolution measurements in the confocal, STED, and STED + deconvolution images obtained by the Leica TCS SP8 STED 3X confocal microscope were also done according to Sparrow [41].

## Figures and Tables

**Figure 1 ijms-22-01903-f001:**
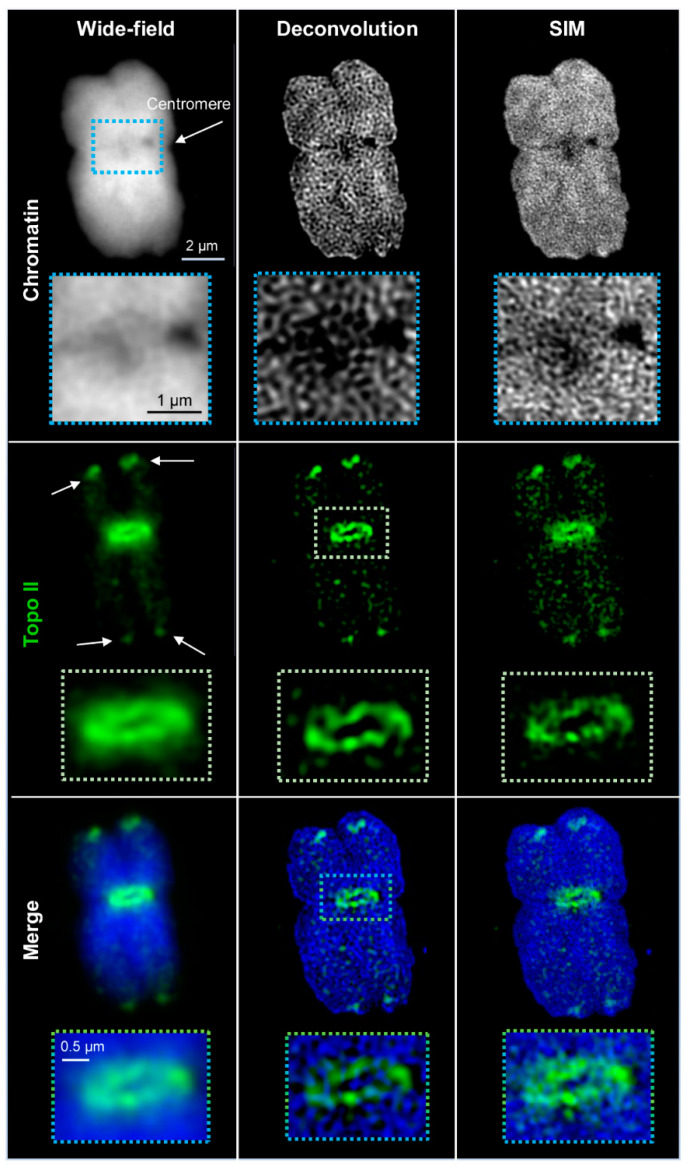
Distribution of topoisomerase II (Topo II) in barley metaphase chromosomes visualized by wide-field, deconvolution, and structured illumination microscopy (SIM) using Alexa488-labeled secondary antibodies. Topo II is accumulated at the subtelomeres (arrows) and the centromere by surrounding it. The enlarged regions (dashed rectangles) show clearly the improved resolution of Topo II and DAPI (4′,6-diamidino-2-phenylindol)-labelled chromatin structures via SIM. Only one representative slice from the Z image stack is shown in all images.

**Figure 2 ijms-22-01903-f002:**
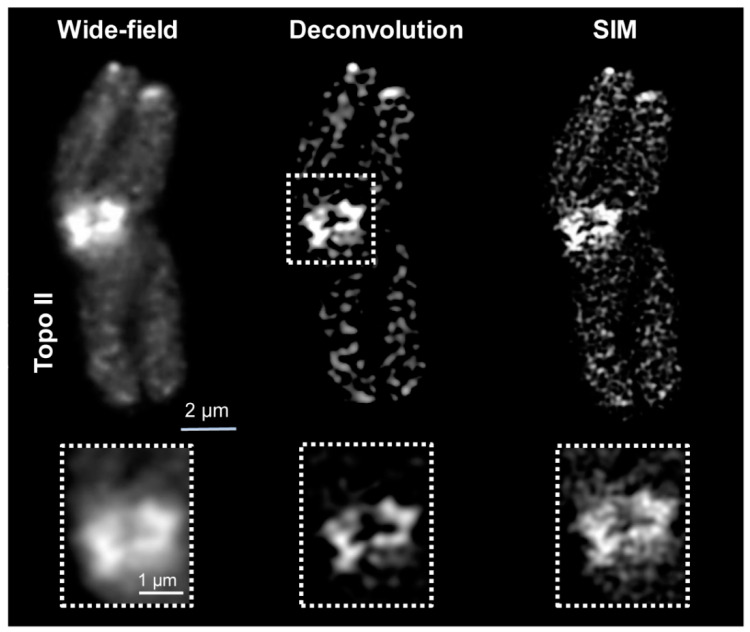
Distribution of Topo II visualized by wide-field, deconvolution, and SIM using STAR635P-labeled secondary antibodies designated for STED. The enlarged regions (dashed rectangles) show clearly the improved resolution achieved via SIM. Only one representative slice from the Z image stack is shown in all images.

**Figure 3 ijms-22-01903-f003:**
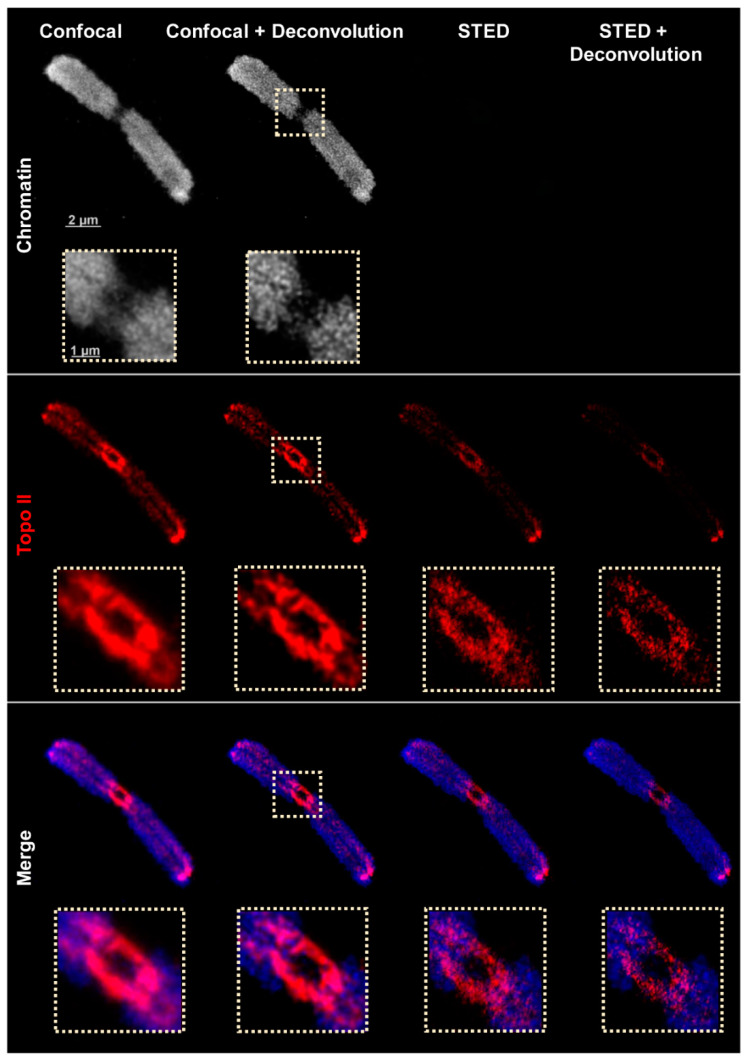
Distribution of Topo II visualized by confocal, confocal + deconvolution, and stimulated emission depletion (STED) + deconvolution using STAR635P-labeled secondary antibodies designated for STED. The enlarged regions (dashed rectangles) show clearly the improved resolution achieved via STED. Only one representative slice from the Z image stack is shown in all images. The STED images were merged with confocal DAPI-labeled chromatin images (blue).

**Figure 4 ijms-22-01903-f004:**
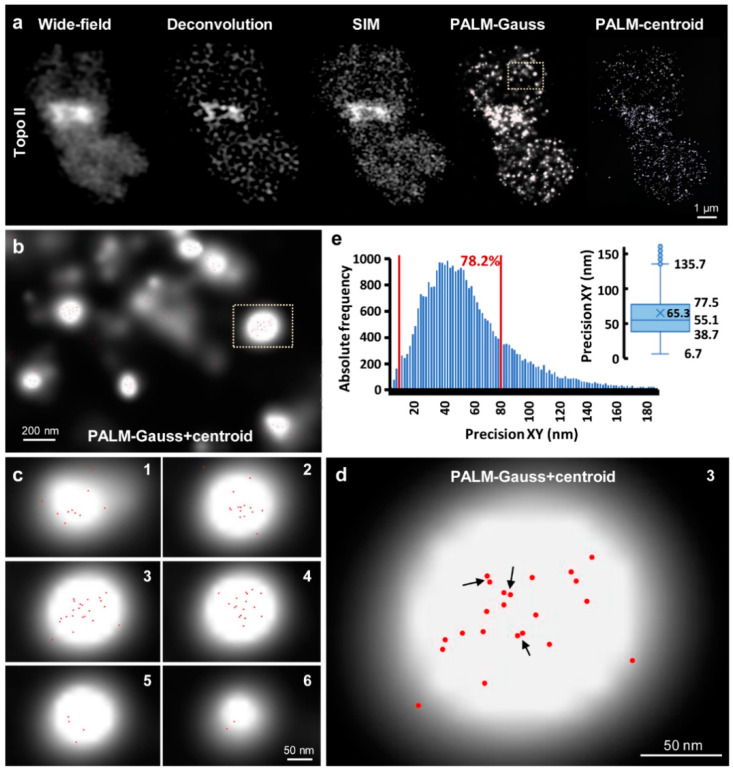
Photoactivated localization microscopy (PALM) allows detecting single Topo II molecules with an XY-precision of up to ~5 nm. (**a**) Visualization of Topo II by wide-field, deconvolution, SIM, and PALM using secondary Alexa488-labeled antibodies in a single 3D image stack slice. The PALM-centroid view indicates the localization of single molecules present in a single PALM image slice of 20 nm thickness. (**b**) Enlarged region (dashed rectangle in (**a**) showing the distinct accumulation of several molecules (indicated by centroids) within PALM-Gauss spots. (**c**) In the six consecutive slices of a further enlarged spot (dashed rectangle in (**b**) differently localized molecules per slice appear. The Z precision reached up to ~45–50 nm (see Appendix A). (**d**) In slice 3 centroid pairs even if ~5 nm apart (arrows) can be distinguished and hence Topo II molecules can be counted. (**e**) The diagram shows the 3D-PALM XY-localization precision of all molecules detected in the whole chromosome. 78.2% of the molecules were localized with a precision of 10–80 nm (region within both red lines). The inset shows the distribution (boxplot) of the lateral localization precision. Numbers indicate lower whisker (6.7 nm), 25% quantile (38.7 nm), median (55.1 nm), mean (65.3 nm), 75% quantile (77.5 nm), and upper whisker (135.7 nm). Localization precisions smaller than 5 nm occurred but were rare.

**Figure 5 ijms-22-01903-f005:**
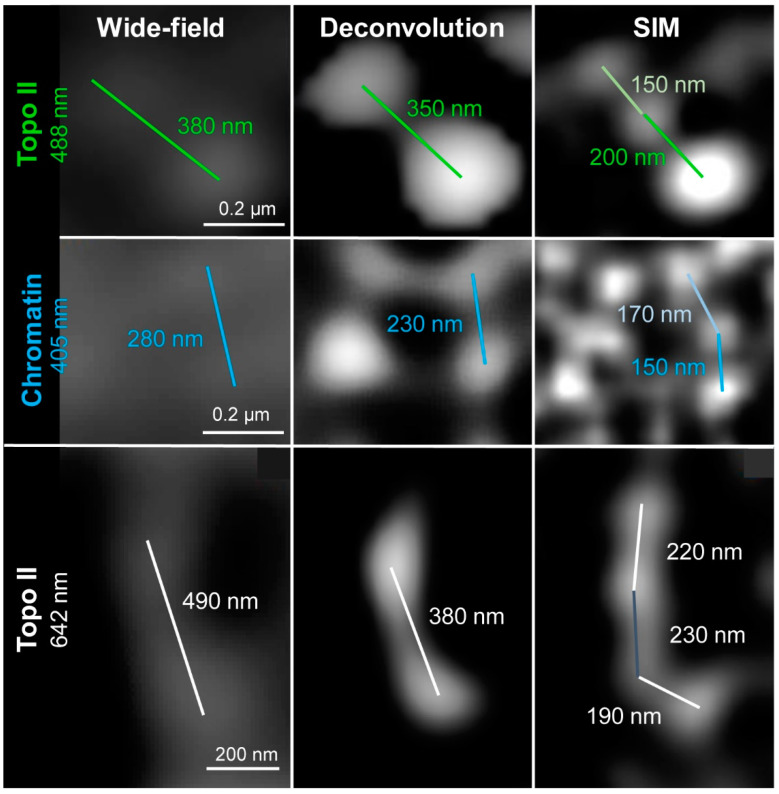
Measurements of the maximal achieved resolution of wide-field, deconvolved wide-field, and SIM by applying different fluorochromes. All images were acquired using the Elyra PS.1 microscope system. The achieved resolution was measured as the distance between the centers of two recognizable spots. Excitation laser lines were 488 nm for Topo II (**upper** panel), 405 nm for chromatin (**middle** panel), and 642 nm for Topo II (**lower** panel). SIM revealed that bigger spots detected in wide-field and deconvolution consist of subclusters.

**Table 1 ijms-22-01903-t001:** Comparison of practical resolutions (in nm) of wide-field (WF), confocal laser scanning microscopy CLSM, structured illumination microscopy (SIM), stimulated-emission depletion (STED), and single-molecule localization microscopy (SMLM).

Microscopy	Lateral Resolution	Axial Resolution	Platform	Fluorochrome Requirement
WF	~200–300	~500–700	Wide-field	Standard
CLSM	~150–220	~360–500	Confocal	Standard
SIM	~100–120	~350–400	Wide-field	Standard
STED	~70–90 *	100–200 **	Confocal	Photostable
PALM	~20–50	~10–70	Wide-field	Switchable

For STED, specific dyes that are more resistant to bleaching, resolutions up to ~50 * and ~90 ** nm can be achieved.

**Table 2 ijms-22-01903-t002:** Maximal achieved lateral resolutions in wide-field, deconvolution, and SIM images after applying different fluorescence dyes in comparison to the theoretically possible resolutions. The numbers of measured chromosomes are in parentheses.

Achieved Resolution	DAPI (nm) λ_em_ = 461 nm	Alexa488 (nm) λ_em_ = 520 nm	STAR635P (nm) λ_em_ = 651 nm
**Wide-field**	297 ± 11 (6)	421 ± 18 (6)	499 ± 22 (7)
**Deconvolution**	216 ± 6 (6)	303 ± 11 (6)	363 ± 9 (7)
**SIM**	131 ± 9 (6)	171 ± 7 (6)	234 ± 21 (7)
**Theoretical Resolution w/NA 1.4**			
**WF (Sparrow)**	~160	~180	~220
**DCV (WF/√2)**	~110	~130	~155
**SIM (WF/2)**	~80	~90	~110
**WF (Raleigh)**	~200	~230	~290
**DCV (WF/√2)**	~140	~165	~205
**SIM (WF/2)**	~100	~115	~145

**Table 3 ijms-22-01903-t003:** Mean maximal achieved resolution in confocal, deconvolution, STED, and STED + deconvolution images after applying secondary STAR635P anti-Topo II antibodies in different embedding media (DABCO and Diamond). The number of measured chromosomes is in parentheses.

Resolution	DABCO (nm)	Diamond (nm)
Confocal	468 ± 38 (6)	432 ± 36 (6)
Deconvolution	243 ± 22 (6)	237 ± 23 (6)
STED	138 ± 23 (6)	134 ± 17 (6)
STED + Deconvolution	78 ± 15 (6)	72 ± 12 (6)

**Table 4 ijms-22-01903-t004:** Resolution criteria and obtainable resolutions for fluorescence wide-field imaging.

Criterium	Lateral Resolution	Axial Resolution
Raleigh	$r_{lat}^{Raleigh}=0.61\cdot \frac{\lambda_{em}}{NA}$	$r_{ax}^{Raleigh}=2\cdot \frac{\lambda_{em}}{NA^{2}}$
FWHM	$r_{lat}^{FWHM}=0.51\cdot \frac{\lambda_{em}}{NA}$	$r_{ax}^{FWHM}=\frac{0.88\cdot \lambda_{em}}{n-\sqrt{n^{2}-NA^{2}}}$
Sparrow	$r_{lat}^{Sparrow}=0.47\cdot \frac{\lambda_{em}}{NA}$	$r_{ax}^{Sparrow}=1.7\cdot \frac{\lambda_{em}}{NA^{2}}$

Parameters: *r_l_*_at_ = lateral resolution, *r_ax_* = axial resolution, *λ_em_* = emission wavelength, NA = numerical aperture of the objective. *n* = refractive index of medium.

**Table 5 ijms-22-01903-t005:** Resolution improvement for confocal and super-resolution techniques.

Method	Improvement Factor	Parameter
DCV	$r^{DCV}=\frac{r^{conv}}{\sqrt{2}}$	Fixed number
CLSM	$r^{CLSM}=\frac{r^{conv}}{\sqrt{2}}$	Fixed number
STED	$r^{STED}=\frac{r^{conv}}{g\cdot \sqrt{1+\frac{I}{I_{s}}}}$	*I* = intracavity intensity *I_sat_* = saturation intensity *g* = geometric factor
SIM	$r^{SIM}=\frac{r^{conv}}{2}$	Fixed number
SMLM	$r^{SMLM}=\frac{r^{conv}}{g\cdot \sqrt{N}}$	*N* = number of photons *g* = geometric factor

Parameters: *r^conv^* = resolution (lateral or axial) of conventional wide-field microscopy.

**Table 6 ijms-22-01903-t006:** Beam widths and associated intensities.

Beam Width	Definition	% of Maximum
Sparrow	$w^{Sparrow}=\frac{0.41}{0.51}\cdot 2\cdot \sqrt{2\cdot ln2}\cdot \sigma \approx 0.3414\cdot \sigma$	98.6
SD	$\sigma =\frac{0.51}{2\cdot \sqrt{2\cdot ln2}}\cdot \frac{\lambda }{NA}\approx 0.2166\cdot \frac{\lambda }{NA}$	60.7
FWHM	$w^{FWHM}=2\cdot \sqrt{2\cdot ln2}\cdot \sigma \approx 2.3548\cdot \sigma$	50
Raleigh	$w^{Raleigh}=\frac{0.61}{0.51}\cdot 2\cdot \sqrt{2\cdot ln2}\cdot \sigma \approx 2.8165\cdot \sigma$	37

Parameter: SD = standard deviation of Gauss function.

## Data Availability

The data that support the findings of this study are available from the corresponding author upon reasonable request.

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
