# Peer review of "Comparing Super-Resolution Microscopy Techniques to Analyze Chromosomes"

_ijms, 2021, doi:10.3390/ijms22041903_

Round 1

Reviewer 1 Report

The development of super-resolution fluorescence microscopy has increased the resolution down to the ~10 - 100 nm range and thus defined a new state-of-the-art in bioimaging to obtain deeper insights into cellular/nuclear structures.

The authors present data of immuno-detected topoisomerase II (Topo II) on DAPI stained mitotic barley chromosomes using different super-resolution microscopic approaches, namely the wide-filed Zeiss Elyra for SIM and PALM, and the LeicaTCS SP8 for STED.  

Introduction:

Authors should at least briefly mention what is known (if anything at all) on the function of Topo II in mitotic plant chromosomes and why they chose this enzyme for their investigation.

Authors provide in Table 1 data for the lateral/axial resolution of the different SR approaches used. Please include the respective references for resolution limits in this table and take care that cited references include data on axial resolution as well.

Results / Mat & Meth:

All legends should be outlined in more detail: Are images shown in Figs. 1-3 for SIM and STED single sections? Please provide information on the z-distance between sections in case of image stacks.

Please provide information on the parameters used for deconvolution.

Fig. 4: For the very small distances <5nm: were multiple blinking events excluded in these cases?

The movies should contain more information. Scale bars, number of sections, image processing. The 4 links supplied afterwards do not even provide information to which of the 3 movies listed in supplementary material they belong.

Fig. 5: distances should read “nm” instead of “µm”. How were the (gravity Intensity?) centers of two recognizable spots defined? How was an individual spot defined?

The formula in section 4.7. contains apparently erroneous characters (l is replaced by ??; explain ‘a’)

General remarks:

As the authors point out in the discussion, the choice of the best/(most appropriate  microscopic approach or combination of different approaches depends on the biological specimen and the particular question addressed. Overall, the content of this study would of course have gained a lot by providing at least some ideas of a gain of biological insight with increased resolution of Topo II substructures.  

As the paper is paper is fully focused on the technical/methodological aspects of resolution assessment, a more detailed description of these approaches should be provided.

Author Response

Dear Reviewer 1,

thank you very much for the helpful comments to improve our manuscript. All changes/additions are highlighted in the revised main text and supplementary word files.

Comments and Suggestions for Authors

The development of super-resolution fluorescence microscopy has increased the resolution down to the ~10 - 100 nm range and thus defined a new state-of-the-art in bioimaging to obtain deeper insights into cellular/nuclear structures.

The authors present data of immuno-detected topoisomerase II (Topo II) on DAPI stained mitotic barley chromosomes using different super-resolution microscopic approaches, namely the wide-filed Zeiss Elyra for SIM and PALM, and the LeicaTCS SP8 for STED.  

Introduction:

Authors should at least briefly mention what is known (if anything at all) on the function of Topo II in mitotic plant chromosomes and why they chose this enzyme for their investigation.

Answer: We wrote in the introduction: “In plants, Topo II is involved in cell cycle regulation as demonstrated for onion [28] and tobacco [29]. Moreover, plant Topo II acts in meiosis to remove bivalent interlocks in Arabidopsis [30].”

And we added: “However, nothing is known about the distribution and function of Topo II in mitotic plant chromosomes. Therefore, we chose this enzyme for our investigations.”

Authors provide in Table 1 data for the lateral/axial resolution of the different SR approaches used. Please include the respective references for resolution limits in this table and take care that cited references include data on axial resolution as well.

Answer: We added a chapter “4.7. Resolution considerations” where we discuss theoretical and practical resolutions and adjusted table 1 accordingly.

Results / Mat & Meth:

All legends should be outlined in more detail: Are images shown in Figs. 1-3 for SIM and STED single sections? Please provide information on the z-distance between sections in case of image stacks.

Answer: We added the information into legends 1-3: “Only one representative slice from the Z image stack is shown in all images.” The information on the z-distance between sections is added to 4.5. and 4.6. of M&M.

Please provide information on the parameters used for deconvolution.

Answer: For STED image deconvolution the parameters were added to 4.6.

For SIM reconstruction Wiener deconvolution was used as implemented in the Zeiss SIM module (Karras, et al. Successful optimization of reconstruction parameters in structured illumination microscopy – A practical guide. 2019. Optics Communications 436: 69-75). The Wiener parameter was first determined automatically and then changed manually to a value where noise structuring was just not visible (see 4.5).

Fig. 4: For the very small distances <5nm: were multiple blinking events excluded in these cases?

Answer: Multiple blinking events were not discarded but accounted for in the fit algorithm. The image just shows localizations of centroids as determined by Gauss fitting. Please note, that the peak to peak distance between centroids is neither the localization precision nor the resolution. The former is determined by the standard deviation of the Gauss fit; the latter is equal to the former if molecule distance is at least half the localization precision, otherwise twice this distance. Please note that label distances are normally not known.

The movies should contain more information. Scale bars, number of sections, image processing. The 4 links supplied afterwards do not even provide information to which of the 3 movies listed in supplementary material they belong.

Answer: In the supplementary material file the movies 1-4 are described. All of them were produced from the chromosome shown and described in figure 4 (including bar size). Scale bars are also visible in movies 2 and 4. In 4.5. of M&M is written: “3D rendering of SIM and PALM image stacks to produce movies was performed with the Imaris 9.6 software (Bitplane). The number of slices used is indicated for SIM and PALM rendering in the movie legends with 14 and 71 slices, respectively.

The information on the z-distance between sections is added to 4.5. of M&M.

Fig. 5: distances should read “nm” instead of “µm”.

Answer: Thank you for the hint. We corrected.

How were the (gravity Intensity?) centers of two recognizable spots defined? How was an individual spot defined?

Answer: The intensity centers were detected visually by choosing the highest spot brightness.

Just to add to this, that this is correct. You can draw a profile over the structure and strictly you would have to fit the curve to a Gauss to determine the mass of gravity. But as an approximation it is the brightest spot.

In localization microscopy gravity intensities are determined by Gauss fitting based on the localization precision measurements. They come with uncertainty, the standard deviation s of the Gauss fit. s determines the maximum achievable resolution.

The formula in section 4.7. contains apparently erroneous characters (l is replaced by ??; explain ‘a’)

Answer: We corrected in the text. It’s now included in a more comprehensive paragraph.

General remarks:

As the authors point out in the discussion, the choice of the best/(most appropriate  microscopic approach or combination of different approaches depends on the biological specimen and the particular question addressed. Overall, the content of this study would of course have gained a lot by providing at least some ideas of a gain of biological insight with increased resolution of Topo II substructures.  As the paper is paper is fully focused on the technical/methodological aspects of resolution assessment, a more detailed description of these approaches should be provided.

Answer: We extended the information about the super-resolution techniques, including now also Airyscan, especially in the introduction.

Submission Date 22 December 2020

Date of this review 12 Jan 2021 23:13:37

Reviewer 2 Report

This is an excellent manuscript comparing WF, Decon and SIM for imaging chromosomes and the authors did a great job in presenting as well. I have the following suggestion, should be considered positively will enhance I believe the quality of manuscript further.

In the Introduction, it would be great to discuss why the Airyscan modality is not tested together with the other techniques. Though the resolution in Airyscan is slightly lower than SIM, the signal-to-noise ratio and as it is less prone to artefacts such as SIM technique would have yielded a valuable comparison. The main reason for this is by seeing Table 1, each technique used for comparison are different in terms of fluorophore requirement, had the authors included the Airyscan data, it would have provided an another technique where standard fluorophores could have been used which will make the SIM and Airyscan technique comparison as 'apples-to-apples' comparison. As of now the techniques compared only based on resolution performance, but each operates in different principle.

Speaking on artefacts, the authors did not provide details of the deconvolution setting used to process the images, this is a must to understand, whether a blind deconvolution routine or an experimental PSF has been used to deconvolve and the number of iterations so on. I understand, the ZEN SIM module does the decon in background, and since one of the authors is from Zeiss, they could shed more light on the protocol used in the background. 

This is important as you see in Fig. 1 center panel deconvolved images, there is a 'honeycomb' shaped artefacts are apparent which is not there in the raw data as well as this is present in the SIM images as well. The Noise/Wiener filter setting used for Decon and SIM if authors could provide that would be helpful. In addition, whether the SIM data is processed in 'background cut' mode, in which the program removes automatically the background, and it might also remove some valuable data along with. If the authors could provide a 'background shift' processed image as a supplement data for the same data of SIM images in Fig. 1, would be very helpful for the readers.

In the Fig. 3 if the authors could provide a panel from WF Raw and WF deconvolved will be essential to understand, when we use photon-lossy techniques such as confocal combined with deconvolution Vs a technique all the photons are collected such as in WF, and compare that with deconvolved WF, will yield realistic comparison between techniques.

The resolution measurement equations needs to be fixed in section 4.7

Finally in Fig 5 the authors choose to present the actual data with center to center distances to measure the resolution. But it is important for the authors to show FWHM using line profile analysis for each figure and measured FWHM measurements. Providing this info in both lateral and axial dimensions would be great for fundamental understanding of performance of these systems under the hands of authors.

The following references should be cited and discussed in the Intro and discussion in the context of history of SIM techniques and performance comparison as in this manuscript.

https://doi.org/10.1046/j.1365-2818.2000.00710.x

DOI: 10.1038/s41556-018-0251-8

https://doi.org/10.1002/jemt.22732

Thank you for the excellent focused work on Chromosome, with these modifications, I trust, it will become a valuable contribution to the scientific community.

Author Response

Dear Reviewer 2,

thank you very much for the helpful comments to improve our manuscript. All changes/additions are highlighted in the revised main text and supplementary word files.

The manuscript by Kubalova et. al. reports on comparative application of three microscopic imaging methods such as wide field imaging (WF), structured illumination (SIM), stimulated emission depletion (STED) and single-molecule localization method (PALM) to plant research, specifically chromatin and topoisomerase II (Topo II), which is a protein important for correct chromatin condensation.

While the study overall can be of interest to certain readers, I have a major problem with the results obtained by PALM. 
Figures 1 (WF and SIM with green dye), 2 (WF and SIM with red dye) and 3 (STED with red dye),  are in general consistent to each other, despite certain differences is resolution - one can see a ring formed around the centromere, which kind of makes sense. 
Figure 4, recorded by PALM, is strikingly different. There is not even a hint of the ring structure, nicely resolved by STED in Figure 3 or by SIM on the same sample in Figure 4a. At the same time obtained resolutions reported for PALM are out of this world - see the next comment. 

Answer: The higher the localization precision, the higher is the intensity for a molecule displayed in the localization map. Hence, intensity is related not only to numbers of events but also localization precision. So, it can well be, if you have a ring-like structure with many molecules in the ring and few distributed evenly in the center that in the case of SMLM the center is as bright as the ring if molecules are localized to the same precision. This is not the case for WF, SIM, and STED where intensities are related to molecule numbers. So, in this case, brightness and contrast can be set that a strong signal is still displayed, weak ones not.  SMLM brightness and contrast cannot be used in such way. If you look close for the SIM image, you will notice a background staining in the center of the ring, and of course this will show up in SMLM as bright spots if localization precision is high. Also, if you look closely, the ring is actually not a fully closed ring. It only looks that way in WF, STED, and SIM as the resolution is obviously not able to reveal the gaps between the chromatids. If you look, however closer, you will see suggestively that the ring is not closed. And this is more pronounced in SMLM.

We wrote in text: “It has to be emphasized that the ring-like centromeric structure appearing to be mainly closed in WF, DCV, SIM, and STED occurs more open in the PALM localization map. We consider this as real because in the DCV and SIM images such a gap is indicated as well (Figure 4a).”

The sentence "The maximal XY-precision for two neighboring molecules within the cluster was measured with ~5 nm." is not sufficiently backed up by the experimental data and methods. Although ultra-high resolutions down to ~10 nm were demonstrated in methods such as DNA-PAINT and similar (mostly from groups of Ralf Jungmann and Peng Yin) - this required hours of acquisition for accumulation billions of photons emitted from multiple binding events with different fluorophores, plus extensive drift correction using fiducial markers.
The centroid analysis method, used by the authors, is not sufficiently described and no controls for such high resolution are presented.

Answer: We believe there is a mis-phrasing. The centroid’s localization is due to the Gauss fit and the distance between two centroids is in no way related to localization precision nor resolution which is determined by the standard deviation (s) for localization precision, and localization precision as well as labeling density for resolution. Localization precision reflects the maximum possible resolution, which can only be achieved if molecules are spaced twice as fine as the localization precision. Otherwise, the resolution is just twice the spacing.

On lines 233-235 the authors state that "For STED microscopy chromatin counterstaining with e.g., DAPI and Hoechst 33342 cannot be applied." which gives an impression that it's not possible to image chromatin with STED. At the same time, the authors cite reference 16, a review by LukinaviÄŤius, which mentions at least 6 dyes directly interacting with DNA which would be compatible with the STED instrument used by the authors.

Answer: We used DAPI for sorting and DNA counterstaining, which is good for SIM.  Our STED (with far red laser) has the best resolution in the far red spectrum. Therefore, we used for Topo II red labeling, to get the best results. But chromatin for STED scanning could be labeled e.g by SPirochrome. We wrote in discussion: “For STED, dyes shifted towards the red spectrum are generally preferred to minimize autofluorescence during imaging. Therefore, dyes with fluorescence excitation within the blue and green spectral ranges are avoided. To achieve best resolved chromatin with low background, Spirochrome’s SiR probes emitting light in the near infrared spectral range are recommended.”

Some other statements could be better phrased, such as lines 223-225: "According to electron microscopy imaging, the diameter of a Topo II dimer is ~15 nm [32], thus only PALM can recognize such single molecules labeled by fluorescence dyes." First, it would be better to replace PALM with more general reference to SMLM - single molecule localization methods. Second, it would be appropriate to mention MINFLUX imaging method, which was shown to obtain single-digit resolution and is available commercially. Third, the authors need to take into account the size of their antibody/antibodies and their flexibility - this would increase the size of the imaged structure. For reference, see 'nanobodies for SMLM'.

Answer: We rephrased: “According to electron microscopy imaging, the diameter of a Topo II dimer is ~15 nm [32], thus in praxis only SMLM would be able to separate such clusters if in such a close proximity.” In principle, all non-linear methods like non-linear SIM, STED, and SMLM would be able to resolve such structures, but because the photon budget is limited in practical terms only SMLM would be able to resolve to 15 nm. The more recent introduced MINFLUX would even be more appropriate with its absolute resolution down to atomic size. Despite this possibility, there are two limitations. First for a certain resolution to be achieved the labels must be spaced twice as fine (Nyquist criterion) and hence the size of the label will matter. Assume antibodies 10 nm in size binding opposite sites to a structure, their labels might be 20 nm apart and hence the maximum resolution would be 40 nm. Second, the photon budget in a biological system always will be limited and hence practical maximal resolutions exist for all super-resolution techniques.

Submission Date 22 December 2020

Date of this review 11 Jan 2021 02:01:32

Reviewer 3 Report

The manuscript by Kubalova et. al. reports on comparative application of three microscopic imaging methods such as wide field imaging (WF), structured illumination (SIM), stimulated emission depletion (STED) and single-molecule localization method (PALM) to plant research, specifically chromatin and topoisomerase II (Topo II), which is a protein important for correct chromatin condensation.

While the study overallcan be of interest to certain readers, I have a major problem with the results obtained by PALM. 
Figures 1 (WF and SIM with green dye), 2 (WF and SIM with red dye) and 3 (STED with red dye),  are in general consistent to each other, despite certain differences is resolution - one can see a ring formed around the centromere, which kind of makes sense. 
Figure 4, recorded by PALM, is strikingly different. There is not even a hint of the ring structure, nicely resolved by STED in Figure 3 or by SIM on the same sample in Figure 4a. At the same time obtained resolutions reported for PALM are out of this world - see the next comment. 

The sentence "The maximal XY-precision for two neighboring molecules within the cluster was measured with ~5 nm." is not sufficiently backed up by the experimental data and methods. Although ultra-high resolutions down to ~10 nm were demonstrated in methods such as DNA-PAINT and similar (mostly from groups of Ralf Jungmann and Peng Yin) - this required hours of acquisition for accumulation billions of photons emitted from multiple binding events with different fluorophores, plus extensive drift correction using fiducial markers.
The centroid analysis method, used by the authors, is not sufficiently described and no controls for such high resolution are presented.

On lines 233-235 the authors state that "For STED microscopy chromatin counterstaining with e.g., DAPI and Hoechst 33342 cannot be applied." which gives an impression that it's not possible to image chromatin with STED. At the same time, the authors cite reference 16, a review by LukinaviÄŤius, which mentions at least 6 dyes directly interacting with DNA which would be compatible with the STED instrument used by the authors.

Some other statements could be better phrased, such as lines 223-225: "According to electron microscopy imaging, the diameter of a Topo II dimer is ~15 nm [32], thus only PALM can recognize such single molecules labeled by fluorescence dyes." First, it would be better to replace PALM with more general reference to SMLM - single molecule localization methods. Second, it would be appropriate to mention MINFLUX imaging method, which was shown to obtain single-digit resolution and is available commercially. Third, the authors need to take into account the size of their antibody/antibodies and their flexibility - this would increase the size of the imaged structure. For reference, see 'nanobodies for SMLM'.

Author Response

Dear Reviewer 3,

thank you very much for the helpful comments to improve our manuscript. All changes/additions are highlighted in the revised main text and supplementary word files.

This is an excellent manuscript comparing WF, Decon and SIM for imaging chromosomes and the authors did a great job in presenting as well. I have the following suggestion, should be considered positively will enhance I believe the quality of manuscript further.

In the Introduction, it would be great to discuss why the Airyscan modality is not tested together with the other techniques. Though the resolution in Airyscan is slightly lower than SIM, the signal-to-noise ratio and as it is less prone to artefacts such as SIM technique would have yielded a valuable comparison. The main reason for this is by seeing Table 1, each technique used for comparison are different in terms of fluorophore requirement, had the authors included the Airyscan data, it would have provided an another technique where standard fluorophores could have been used which will make the SIM and Airyscan technique comparison as 'apples-to-apples' comparison. As of now the techniques compared only based on resolution performance, but each operates in different principle.

Answer: We added a section in the introduction to explain why we have omitted Airyscan. In short, Airyscan would have yielded a similar resolution indeed to SIM, but not better. As a confocal technique it would normally be superior when working with thicker samples. But with our chromosome spreads, there would have been no advantage over SIM. There is a misconception as to signal-to-noise (SNR). Albeit in Airyscan the pinhole can be opened to improve the SNR over a conventional confocal microscope with a closed pinhole, WF has still a much better SNR. Only when sectioning is required to block out-of-focus light and for high scattering samples SNR of a confocal microscope might outmatch the one from WF.

Speaking on artefacts, the authors did not provide details of the deconvolution setting used to process the images, this is a must to understand, whether a blind deconvolution routine or an experimental PSF has been used to deconvolve and the number of iterations so on. I understand, the ZEN SIM module does the decon in background, and since one of the authors is from Zeiss, they could shed more light on the protocol used in the background. 

Answer: In the ZEN SIM reconstruction SW Wiener deconvolution is used. As a matter of fact, Wiener deconvolution is not in the strict sense a deconvolution but rather a regularization, i. e. the image is regularized by dividing through the Wiener factor. No iterative step is involved and Wiener filtering is therefore as a linear method in a strict sense the only quantitative “deconvolution” compared to iterative approaches. The downside is that iterative approaches will yield higher resolutions but at the same time are more prone for artifacts or yielding the wrong structure.

This is important as you see in Fig. 1 center panel deconvolved images, there is a 'honeycomb' shaped artefacts are apparent which is not there in the raw data as well as this is present in the SIM images as well. The Noise/Wiener filter setting used for Decon and SIM if authors could provide that would be helpful. In addition, whether the SIM data is processed in 'background cut' mode, in which the program removes automatically the background, and it might also remove some valuable data along with. If the authors could provide a 'background shift' processed image as a supplement data for the same data of SIM images in Fig. 1, would be very helpful for the readers.

Answer: First, we did SIM reconstruction with a free Wiener parameter and then manually changed its value to a setting, where noise artifacts (referred to as hammerhead or honeycomb) just were not visible. We do not interpret the center panel structures as honeycomb; they stem from true signal. As DCV and SIM reassign photons to create sharp edges, the images will be of higher contrast. As a matter of fact there is signal in the widefield (or raw) image; only the contrast in the image is lower. Like in the upper panel, there are brighter and darker parts and those will be deconvolved to brighter and darker ones.

In the Fig. 3 if the authors could provide a panel from WF Raw and WF deconvolved will be essential to understand, when we use photon-lossy techniques such as confocal combined with deconvolution Vs a technique all the photons are collected such as in WF, and compare that with deconvolved WF, will yield realistic comparison between techniques.

Answer: It is not possible to obtain WF with our Leica STED system, because it is based on confocal scanning. WF examples are shown in figs. 1 and 2 acquired with the Zeiss Elyra.

The resolution measurement equations needs to be fixed in section 4.7

Answer: We prepared now a more comprehensive paragraph on resolution in the M&M section.

Finally in Fig 5 the authors choose to present the actual data with center to center distances to measure the resolution. But it is important for the authors to show FWHM using line profile analysis for each figure and measured FWHM measurements. Providing this info in both lateral and axial dimensions would be great for fundamental understanding of performance of these systems under the hands of authors.

Answer: The intensity centers were detected visually by choosing the highest spot brightness according to Sparrow. In fig. S6 and 4.7 and 4.8, we describe now the measurements and resolution criteria more in detail.

The following references should be cited and discussed in the Intro and discussion in the context of history of SIM techniques and performance comparison as in this manuscript.

Gustaffson (2000) Surpassing the lateral resolution limit by a factor of two using structured illumination microscopy. https://doi.org/10.1046/j.1365-2818.2000.00710.x

Schermelleh, et al. (2019). Super-resolution microscopy demystified. Nature Cell Biology, 21(1), 72–84.DOI: 10.1038/s41556-018-0251-8

Sivaguru et al. (2018) Comparative performance of airyscan and structured illumination superresolution microscopy in the study of the surface texture and 3D shape of pollen.Microscopy Research and Technique 2018; 81: 101-114, https://doi.org/10.1002/jemt.22732

Answer: We cite these publications as requested in the Introduction.

Thank you for the excellent focused work on Chromosome, with these modifications, I trust, it will become a valuable contribution to the scientific community.

Submission Date

22 December 2020

Date of this review

13 Jan 2021 19:43:49

Round 2

Reviewer 1 Report

Authors have addressed my concerns and I recommend the manuscript in its present form for publication

Reviewer 2 Report

Thank you for revising the manuscript with requested information, this will be an important contribution to the field of superresolution optical microscopy.

Reviewer 3 Report

I thank the authors for revising the manuscript and recommend it for the publication.